# Semantic Distillation from Neighborhood for Composed Image Retrieval

### Yifan Wang
Tsinghua Shenzhen International School
Tsinghua University
Shenzhen, China
yifan-wa22@mails.tsinghua.edu.cn

### Lei Li
Tsinghua Shenzhen International School
Tsinghua University
Shenzhen, China
lei-li18@mails.tsinghua.edu.cn

### Wuliang Huang
Institute of Computing Technology, Chinese Academy of Sciences
Beijing, China
huangwuliang19b@ict.ac.cn

### Chun Yuan*
Tsinghua Shenzhen International School
Tsinghua University
Shenzhen, China
yuanc@sz.tsinghua.edu.cn

## Abstract

The challenging task composed image retrieval targets at identifying the matched image from the multi-modal query with a reference image and a textual modifier. Most existing methods are devoted to composing the unified query representations from the query images and texts, yet the distribution gaps between the hybrid-modal query representations and visual target representations are neglected. However, directly incorporating target features on the query may cause ambiguous rankings and poor robustness due to the insufficient exploration of the distinguishments and overfitting issues. To address the above concerns, we propose a novel framework termed *SemAntic Distillation from Neighborhood (SADN)* for composed image retrieval. For mitigating the distribution divergences, we construct neighborhood sampling from the target domain for each query and aggregate neighborhood features with adaptive weights to restructure the query representations. Specifically, the adaptive weights are determined by the collaboration of two individual modules, as correspondence-induced adaption and divergence-based correction. Correspondence-induced adaption accounts for capturing the correlation alignments from neighbor features under the guidance of the positive representations, and the divergence-based correction regulates the weights based on the embedding distances between hard negatives and the query in the latent space. Extensive results and ablation studies on CIRR and FashionIQ validate that the proposed semantic distillation from neighborhood significantly outperforms baseline methods.

## CCS Concepts

• **Information systems** → **Information retrieval**; • **Computing methodologies** → *Visual content-based indexing and retrieval.*

## Keywords

Multi-modal Retrieval, Composed Image Retrieval, Deep Metric Learning

**ACM Reference Format:**
Yifan Wang, Wuliang Huang, Lei Li, and Chun Yuan. 2024. Semantic Distillation from Neighborhood for Composed Image Retrieval. In *Proceedings of the 32nd ACM International Conference on Multimedia (MM '24), October 28-November 1, 2024, Melbourne, VIC, Australia.* ACM, New York, NY, USA, 9 pages. https://doi.org/10.1145/3664647.3681493

## 1 Introduction

Achieving mutual understanding across diverse modalities (such as images, texts, and videos) [4, 7, 23] has long been a fundamental concern in the field of artificial intelligence research. With the remarkable development of social media platforms, the trend towards utilizing multimodal inputs for user queries is gaining momentum for a more convenient and efficient expression of users' requirements. However, traditional image search [45, 49] and text-to-image retrieval [2, 26] could not support this desire to comprehend the multi-modal query directly. The task of Composed Image Retrieval (CIR) [1, 3, 9, 24, 38] naturally arises out of necessity to search for the required images given the input composed of a reference image and a sentence describing the modifications on the query image. Through attaining high-level conceptual comprehension and implicit semantics, researches on the CIR promote the related applications, *e.g.*, vision reasoning [17, 21] and change captioning [36, 37], and drive the growth of cognitive intelligence.

Despite the remarkable advancements [5, 16, 47], this interactive composed image retrieval still poses a significant challenge. The contents in query images and modifiers complement each other for they form a complete user's requirement, yet they also conflict with each other for the textual information implying modifications on images. Drawing upon the above facts, the model involves an effective combination of the semantics to retain and alter, to bring it closer to the target feature within the shared subspace via appropriate measurement metrics. To tackle the aforementioned obstacles, prior approaches can be categorized into two classes, *i.e.*, compositional learning and target-guided learning. The first category [9, 20, 24, 44] resorts to global feature fusion or synthesizes

*Corresponding author.

*MM '24, October 28-November 1, 2024, Melbourne, VIC, Australia*
© 2024 Copyright held by the owner/author(s).
ACM ISBN 979-8-4007-0686-8/24/10
https://doi.org/10.1145/3664647.3681493

local alignment clues from the query images and query texts to obtain the unified representations for the hybrid-modal query. Under the guidance of the target images, the second category [13, 27, 41] incorporates the visual features in the target domain into similarity measurements to achieve sufficient interactions within the semantic triplets *i.e.*, query images, modification sentences, and target images. For example, ARTEMIS [13] decomposes the query-to-target similarity into implicit similarity to compare the correlated triplet and explicit similarity to measure the shared characteristics between query text and the target image, yet the guidance of targets usually entails significant time consumption and computation cost especially in testing.

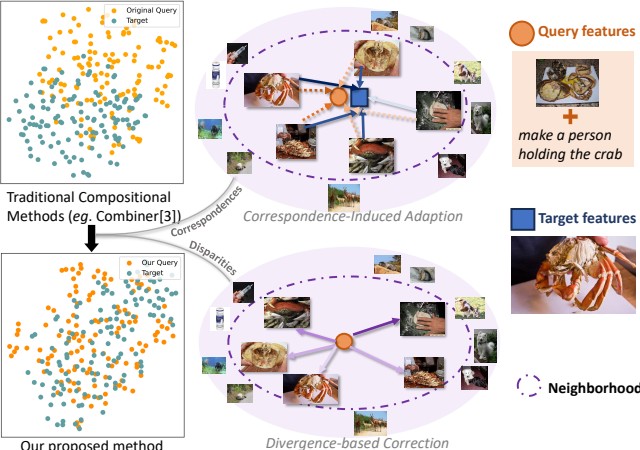

**Figure 1: Illustration of proposed SADN. Distribution divergence exists in previous compositional methods (seen in the top-left corner). After the cooperation of Correspondence-Induced Adaption to extract semantic alignments and Divergence-based Correction to repel irrelevant information, our SADN refines the query features to better align matched embeddings (seen in the bottom-left corner).**

Although the abovementioned methods achieve advancement, several crucial issues have been disregarded in the prior research. Firstly, notable distribution discrepancy as shown in Figure 1 is observed between the distributions of query and target, since the target features originate from the visual modality and the query features are mapped in the mixed-modal composition space. The heterogeneous gap could accumulate statistic errors for the divergent distributions and cause misleading alignments, which further impairs the retrieval performance. Secondly, as the annotators for the modifications mainly concentrate on the differences between one-to-one pairs of the reference images and target images [11, 20], an abundance of candidate images conveying visual concepts that are coherent with the multi-modal query are unlabeled as positive instances. For example, in Figure 1, several pictures showing a man seizing the crab that fits the query demands are regarded as negative instances and punished in traditional compositional methods when they are embedded close to the anchor query embeddings. Considering that most previous work exploits batch-based classification loss as the main optimization objective, these false negative

samples with semantic correlations would misdirect the model to overfit the noises. Thirdly, subtle semantic differences remain to be explored and a robust model should be aware of the distinctions between the query and the negative examples which are similar to the target image. Yet existing frameworks still make ambiguous predictions when facing hard negative examples. As shown in Figure 1, though the images with high ranking results (in the purple dashed circle) resemble the query image and target image to some extent, they conflict with the "holding the crab" or "a person" that the text cues in the query domain require. Overall, the above limitations of distribution discrepancy, noisy supervision and model discrimination entail a systematic approach for this task.

For addressing the above issues, we present a novel method named *SemAntic Distillation from Neighborhood* (SADN) to balance the correlation extractions from false negatives and discrimination awareness from neighborhood for composed image retrieval. Figure 2 displays the whole architecture of the proposed approach, which is comprised of three major processes. After extracting features for the target images and the query with reference image and text modifier, we construct neighborhood sampling from the target domain for each query based on the initial similarity ranking. To alleviate misdirecting effects to drive correlated candidates apart from the query for the noisy annotations, the correspondence-induced adaption is introduced to capture correlations from the neighbor representations with the supervision of target representations. Under the adaption constraint from the query to the target, shared characteristics are strengthened during optimization. For promoting the discriminations between matched instances and hard negatives, we exploit Mahalanobis distance measurement in the mixed-modal space to evaluate the distribution divergences of each neighbor sample from the query. The correspondence-induced adaption and the divergence-based correction jointly compose the weights for the neighbor representations, and samples in the neighborhood are adaptively aggregated on the query features to encourage the query-to-target interactions across different domains. Extensive experiments and ablation studies validate the effectiveness of distilling semantics from the constructed neighborhood on Fashion-IQ [42] and CIRR [29].

In summary, the contributions of the SADN are listed as:

- We propose a novel Semantic Distillation from Neighborhood approach dubbed SADN for composed image retrieval, which incorporates shared semantic characteristics from the targets into query representations to bridge the distribution discrepancy between queries and targets.
- We construct the neighborhood for each query, which captures inherent semantic correlations and filters irrelevant components from the target domain to reconstruct the query features with adaptive weights for improving the robustness of the model.
- We apply correspondence-induced adaption with the guidance of target representations to distill alignments between query and target, in collaboration with divergences measurement to evaluate the distribution disparity for refining similarity ranking.

## 2 Related Works

### 2.1 Composed Image Retrieval

Composed image retrieval [3, 24, 38, 39, 41] targets identifying the hidden intention of user queries based on the reference images and sentences describing modifications. TIRG [38] was the first to propose the paradigm of composing reference images and modifiers and measuring the distances between candidate images and the query compositions, which was further refined by [20, 44]. Specialized in the fashion domain, FashionVLP [15] utilized the prior knowledge from large multi-modal corpora to guide fashion matching in the transformer structure with multiple visual context layers. Anwaar [1] and Kim [22] implemented the composed image retrieval into symmetrical bi-directional matching with an additional reverse retrieval from the target to query to strengthen the supervision through reversible transformations. TG-CIR [41] introduced knowledge distillation from the target-guided teacher module to lead the similarity distributions in the student network without targets. To deal with uncertainty issue by unlabeled candidates, Chen *et al.* [10] designed an uncertainty modeling module to learn the tolerable ranges of features with a regularization to ensure the alignments. In summary, most previous works mainly focus on the combinations of query image and modification sentences, and extracting fine-grained matching components from the query to the target, but they ignore the potential correlations lying in the false negative candidates and discriminations between hard negatives and target samples. Through enhancing the neighborhood representations integrated on the query features, the proposed SADN aims to magnify the differences between highly similar instances while addressing the issue of noisy annotations.

### 2.2 Deep Metric Learning

The goal of deep metric learning is to guarantee that similar inputs are mapped to points close in distance while dissimilar inputs are mapped to points far apart in the embedding space. Due to the high dimensionality of the feature space after deep neural networks, one of the core issues for deep metric learning is the efficient computation of pairwise distances in the hidden space. To address this issue, various approximation techniques have been proposed, including triplet loss functions [33], siamese architectures [25], and contrastive learning [18, 31]. The advances in metric learning have led to significant improvements in a range of applications, including image retrieval [6], representation learning [50], and clustering. Specifically, Lim *et al.* [28] designed a novel hypergraph-based loss function to incorporate multiple semantic tuples into the edges, which improved class-discriminative semantic relation learning for the image classification task. To avoid false negative sampling from the negative examples, adaptive false negative elimination and attraction [43] solved the negative sampling issue by selecting instances based on semantic distances from the reference sentence representations. However, there are still issues unsolved when applying deep metric learning in the downstream tasks including the impacts of outliers and noisy labels. In this work, we exploit a flexible trade-off mechanism to balance correcting matching triplets and mining the hard negative samples to promote the discrimination of the model.

## 3 Methodology

### 3.1 Problem Definition

The task of composed image retrieval could be formulated as minimizing the distances between query embeddings and pairwise target embeddings. To formulate this retrieving process, the reference image, modification sentences, and the candidate images are denoted as $\mathcal{I}_q$, $\mathcal{T}_q$, and $\mathcal{I}_t$, respectively. According to the annotations of the datasets for CIR, $\mathcal{I}_q^i, \mathcal{T}_q^i$ and $\mathcal{I}_t^i$ construct a matched triplet from the query to the target. Note that $\mathcal{I}_q^i$ and $\mathcal{T}_q^i$ are always bound together and the model is expected to find the identical $\mathcal{I}_t^i$ for the $i$-th query $(\mathcal{I}_q^i, \mathcal{T}_q^i)$ during validating and testing. In this work, the images and sentences are encoded through pretrained CLIP visual encoders and textual encoders [32] to obtain the primal representations. The reference image $\mathcal{I}_q^i$, the modification sentence $\mathcal{T}_q^i$ and the target image $\mathcal{I}_t^i$ are embedded as $\boldsymbol{I}_i \in \mathbb{R}^d$, $\boldsymbol{M}_i \in \mathbb{R}^d$ and $\boldsymbol{T}_i \in \mathbb{R}^d$, respectively. Therefore, the goal of this interactive retrieval task could be interpreted as maximizing the similarity measurement between the hybrid-modal query embeddings and the corresponding target embeddings in the latent space, as:

$$\max_{\Theta, \Psi} \kappa(f(\boldsymbol{I}_i, \boldsymbol{M}_i, \Theta), g(\boldsymbol{T}_i, \Psi)), \qquad (1)$$

where $f(\boldsymbol{I}_i, \boldsymbol{M}_i, \Theta)$ refers to composition functions to generate query features for reference image $\boldsymbol{I}_i$ and modification text $\boldsymbol{M}_i$ with learnable parameter $\Theta$. $g(\boldsymbol{T}_i, \Psi)$ is the visual encoder with parameter $\Psi$ for the target image. In this work, we simply use the original target features for measurement, *i.e.*, $g(\boldsymbol{T}_i, \Psi) = \boldsymbol{T}_i$. $\kappa(f(\boldsymbol{I}_i, \boldsymbol{M}_i, \Theta), g(\boldsymbol{T}_j, \Psi))$ denotes the kernel functions to calculate the similarity score between the $i$-th query (composed of $\boldsymbol{I}_i$ and $\boldsymbol{M}_i$) and the $j$-th target. The Eq. 1 is optimized to ensure that $\kappa(f(\boldsymbol{I}_i, \boldsymbol{M}_i, \Theta), g(\boldsymbol{T}_i, \Psi)) > \kappa(f(\boldsymbol{I}_i, \boldsymbol{M}_i, \Theta), g(\boldsymbol{T}_j, \Psi))$, where $j \neq i$.

### 3.2 Neighborhood Construction

Prior models usually design delicate mechanisms to combine features of the reference images and modification texts to obtain the query representations and then rank the distances between the query and target representations in the embedding space. Nevertheless, since the reference images and modification sentences may contain conflict semantics in most cases, we argue that overemphasis on the combination of the multi-modal information in the query could bring redundancy. Since textual information describes the alterations on the query images and an overly complicated fusion scheme could cause more noise and ambiguous semantic learning. In the proposed SADN, we apply a concise compositor to yield the raw query features and calculate the initial similarities between the query and target to select neighbor target instances for each query. The neighborhood construction based on the query-to-target ranking is the prerequisite for fine-grained comparison and measurement in subsequent learning to improve the model discrimination. Besides, the neighbor features selected among the candidate images could mirror the concept of query instances lying in the target image domain and guide the query representation to bridge the distribution differences from the query to the target domain.

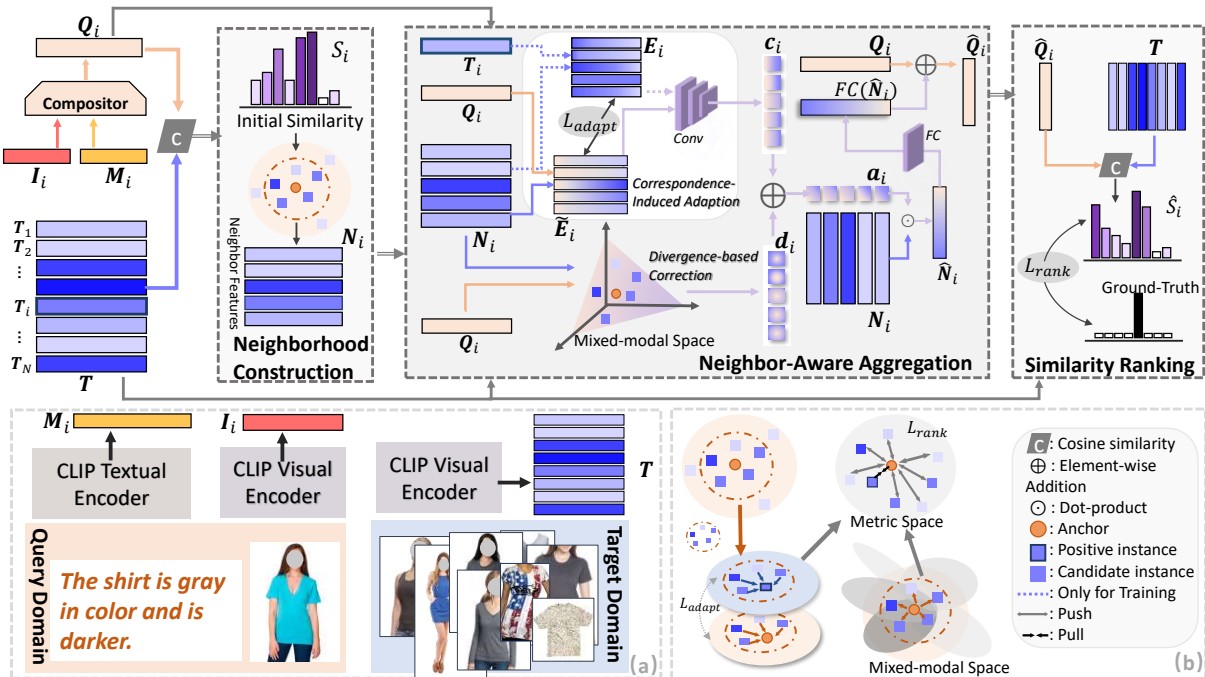

**Figure 2: Overview of the proposed SADN. Given the reference image, modification sentences (in the query domain) and the candidate images (in the target domain), the raw representations are extracted via CLIP encoders, as seen in (a). The neighborhood construction for each query is obtained through ranking the initial similarities $S_i$ between the composed query and targets. The Neighbor-Aware Aggregation is constituted of Correspondence-Induced Adaption to measure the alignments between the neighborhood and the targets, and a Divergence-based Correction to evaluate the distribution divergences between the query and neighbor features. Each candidate sample in the neighborhood with an adaptive weight is aggregated on the query feature, which distills the distribution characteristics from the target domain for the query (See the top box). The transformation of the embeddings of query and candidate examples in the latent space is shown in (b).**

*3.2.1 Multi-modal Compositional Learning.* Given the extracted visual and textual features from the query, an effective compositor to merge the visual information from the reference images and textual semantics from modifiers is essential for obtaining a unified query representation. Since the contributions of the image and text modalities may vary from instances to instances, a weighted visual and textual representation is an intuitive solution to adjust the proportions of different modalities in composing the query features. In this work, we follow the Combiner [3] to integrate the embeddings of concatenation of query image and query text features on the weighted raw features $I_i$ and $M_i$. The composed query feature $Q_i \in \mathbb{R}^d$ could be formulated as:

$$Q_i = Combiner(I_i, M_i) \qquad (2)$$
$$= \alpha MLP_I(I_i) + (1-\alpha)MLP_M(M_i) + FC(\oplus(I_i, M_i)),$$

where $\oplus$ is the concatenation operation and $\alpha$ is the weight controlling the contributions of the visual component in the composed query. $MLP_X$ and $FC$ are the abbreviation of Multi-Layer Perceptron for $X$ modality and fully-connected layer respectively.

*3.2.2 Neighbor Representations.* As introduced above, we construct the neighborhood in the target domain for each query through the ranking of the query-to-target similarities to reflect the query

features and guide the cross-domain alignments. On the basis of the unified query representations $Q_i$, the computation of the initial similarity adopts the widely-used cosine similarity as $s_{ij} = \frac{Q_i^\top \cdot T_j}{\|Q_i\|\|T_j\|}$. For the $i$-th query, we simply use $S_i = \{s_{i1}, s_{i2}, ..., s_{iN}\} \in \mathbb{R}^N$ to refer to the similarity list containing the similarities between the $i$-th query and all the $N$ candidate images in the minibatch. Through ranking the similarities by the descending order, we would screen the top-$K$ candidate images that exhibit a significant degree of similarity with the query as the neighborhood set $\mathcal{S}_i^N = \{j | s_{ij} \in \mathrm{TOP}(s_{ij}, K)\}$. Afterward, the corresponding neighbor representations $N_i \in \mathbb{R}^{K \times d}$ for the $i$-th query are sampling from the visual features in the target domain based on the neighborhood set $\mathcal{S}_i^N$, which is constituted as follows:

$$N_i = \oplus\{g(T_j, \Phi)\}, \quad j \in \mathcal{S}_i^N, \qquad (3)$$

where $\oplus$ is the concatenation operation and $N_i^j \in \mathbb{R}^d$ represents the $j$-th candidate images features in the $i$-th query neighborhood.

### 3.3 Neighbor-Aware Aggregation

The unified query feature $Q_i$ only integrates features from query images and modification texts while it lacks the guidance of the target image to align semantics explicitly. The neighbor features

$N_i$ obtained in Sec. 3.2, originating in the target domain, possess the following characteristics. On one hand, as the neighbor features share high similarity scores with the query, they also exhibit a strong inclination to be false negative samples for the query. Actually, an inherent issue with this interactive task is that multiple candidate images corresponding with the user intents miss annotations to be labeled as positive samples, which could not be neglected. On the other hand, neighbor features inevitably have some nuance to the target features and the retrieval model is expected to be capable of detecting subtle differences and be aware of the real user requirements when dealing with hard negatives. Considering the above two aspects, we design the Neighbor-Aware Aggregation decomposed as two branches, *i.e.*, Correspondence-Induced Adaption to distill correlated semantics from false negatives in the neighborhood, and Divergence-based Correction to percept the distribution differences from hard negatives.

*3.3.1 Correspondence-Induced Adaption.* As aforementioned, this branch prioritizes the guidance from the target features to steer the adaption from the hybrid-modal distributions in the query domain towards visual distributions in the candidate image domain. Specifically, we first establish dual alignments including alignment embeddings of each neighbor sample and target feature $E_i$, and the alignment embeddings of each neighbor sample and query feature $\tilde{E}_i$, to separately measure the resemblances from the neighborhood to the target and query examples, which is calculated via:

$$E_i^j = N_i^j \odot T_i, \quad \tilde{E}_i^j = N_i^j \odot Q_i, \qquad (4)$$

where $\odot$ means pointwise product. We denote $E_i = \{E_i^1, E_i^2, ..., E_i^K\} \in \mathbb{R}^{K \times d}$ and $\tilde{E}_i = \{\tilde{E}_i^1, \tilde{E}_i^2, ..., \tilde{E}_i^K\} \in \mathbb{R}^{K \times d}$ for simplification.

Since the alignment embedding $E_i$ evaluates the correlations between the neighbor instances and target instance, we could derive the correspondence score to infer the correlations between neighbor instance and the $i$-th query among all the neighbors via:

$$c_i^j = \frac{exp(\text{Conv}(E_i^j))}{\sum_j^K exp(\text{Conv}(E_i^j))}, \qquad (5)$$

where Conv denotes a $1 \times 1$ convolutional layer to encode the alignment features into the correspondence score $c_i^j$ for the $j$-th instance in the $i$-th query's neighborhood. A high correspondence score $c_i^j$ implies that the $j$-th candidate image has great semantic overlap with the $i$-th target image. Hence, the $j$-th candidate feature in the $i$-th query neighborhood is supposed to aggregate with more emphasis on the query feature.

Note that in the validating and testing, the guidance of target images is invalid and we substitute the $\tilde{E}_i$ for $E_i$ in Eq. 5. In an ideal form, it behooves the $\tilde{E}_i$ to be consistent with $E_i$, for the alignments between the neighbors with the queries are in parallel with alignments between the neighbors with the targets. We assign a Kullback Leibler (KL) divergence between $\tilde{E}_i$ and $E_i$ during training to ensure the consistency and strengthen the adaptation between query and target domain, which is formulated as:

$$\mathcal{L}_{adapt} = \frac{1}{N} \sum_{i=1}^{N} D_{KL}(\tilde{E}_i \parallel E_i) = \frac{1}{N} \sum_{i=1}^{N} \sum_{j=1}^{K} \tilde{E}_i^j \log \frac{\tilde{E}_i^j}{E_i^j}. \qquad (6)$$

Therefore, with the supervision of the target representations to capture the correlations from neighbors, the query features would be incorporated with shared semantics from false negatives through the dynamic aggregation weights, driving the refined query features and false negative features closer in the latent space.

*3.3.2 Divergence-based Correction.* To enhance the discriminations when comparing highly similar instances among substantial candidate images, the divergence-based correction module is dedicated to the awareness of the distribution divergences from the neighborhood features. From another perspective, neighborhood features in high dimensional latent space could be regarded as nearest feature distributions for the query representations, and precisely depicting the distribution differences could reflect the deviations from the candidate sample to the query. To preserve the inherent distribution characteristics of each representation and measure the intricate distances between high-dimensional representations at a fine level, we apply the squared Mahalanobis distance to measure the neighborhood distributions $N_i^j$ and query features $Q_i$ as:

$$u_i^j = (N_i^j - Q_i)^\top \Sigma^{-1} (N_i^j - Q_i), \qquad (7)$$

$$d_i^j = \frac{exp(1 - u_i^j)}{\sum_j^K exp(1 - u_i^j)}, \qquad (8)$$

where $\Sigma \in \mathbb{R}^{d \times d}$ represents the covariance matrix in the high-dimensional embedding space and is initialized by $\text{diag}(Q_i)$. Therefore, the distance $d_i^j \in [0, 1]$ measures the semantic disparity of neighbor $j$ away from the anchor query $i$.

After distilling the semantic resemblances of the neighborhood from the target and estimate the distribution divergences of the neighborhood to the query, we then integrate the two measurements into one indicator as weight parameters for further aggregation. The adaptive weight $a_i = \{a_i^1, a_i^2, ..., a_i^K\}$ is calculated through:

$$a_i^j = \alpha * c_i^j + \beta * d_i^j, \qquad (9)$$

where $\alpha$ and $\beta$ are learnable parameters.

*3.3.3 Adaptive Aggregation.* After obtaining the neighborhood distributions for each query, it is expected to apply the representations of neighborhood samples to refine the query features. In this way, the distribution characteristics in the target domain could be introduced to guide the query features to be close to the target features in the embedding space, and generalize the query features to adapt to the externalization variety of semantics when comparing the distances to the candidate images. Meanwhile, based on the adaptive weight, the refined query feature could preserve the primitive semantics and distill the correlated semantics adaptively on account of the correspondences and distribution divergences, which could enhance the sensitivities of comparing the true positive instances and hard negative samples. The computation of the adaptive aggregation features is shown as follows:

$$\hat{N}_i = \phi(a_i^j N_i^j), \qquad (10)$$

where $\phi$ represents Sum-pooling to aggregate the $K$ neighbor embeddings as unified representations. Afterwards, we exploit the residual connection to integrate the adaptive aggregation features

**Table 1: Experiments results on FashionIQ. Best and second-best results are marked in bold and underlined respectively.**

| Methods | R@10 | | | | R@50 | | | |
|---|---|---|---|---|---|---|---|---|
| | Dress | Shirt | Toptee | Mean ↑ | Dress | Shirt | Toptee | Mean↑ |
| TIRG (CVPR'19)[38] | 14.87 | 18.26 | 19.08 | 17.40 | 34.66 | 37.89 | 39.62 | 37.39 |
| VAL (CVPR'20)[9] | 21.12 | 21.03 | 25.64 | 22.60 | 42.19 | 43.44 | 49.49 | 45.04 |
| CIRPLANT (ICCV'21)[29] | 14.38 | 13.64 | 16.44 | 14.82 | 34.66 | 33.56 | 38.34 | 35.52 |
| CoSMo (CVPR'21)[24] | 21.39 | 16.90 | 21.32 | 19.87 | 44.45 | 37.49 | 46.02 | 42.65 |
| CLVC-Net (SIGIR'21) [40] | 29.85 | 28.75 | 33.50 | 30.70 | 56.47 | 54.76 | 64.00 | 58.41 |
| ARTEMIS (ICLR'22)[13] | 27.16 | 21.78 | 29.20 | 26.05 | 52.40 | 43.64 | 54.83 | 50.29 |
| MACAM (ACM MM'22)[46] | 30.51 | 33.67 | 30.73 | 31.60 | 57.11 | 64.48 | 58.02 | 59.87 |
| CRN (TIP'23)[44] | 30.20 | 29.17 | 33.70 | 31.02 | 57.15 | 55.03 | 63.91 | 58.70 |
| CSS (Arxiv'23)[48] | 33.65 | 35.96 | 42.65 | 37.42 | 63.16 | 61.96 | 70.70 | 65.27 |
| FashionVLP (CVPR'22 ) [15] | 26.77 | 22.67 | 28.51 | 25.98 | 53.20 | 46.22 | 57.47 | 52.30 |
| CLIP4Cir (CVPR'22)[3] | 31.63 | 36.36 | 38.19 | 35.39 | 56.67 | 58.00 | 62.42 | 59.03 |
| DWC (AAAI'24)[20] | 32.67 | 35.53 | 40.13 | 36.11 | 57.96 | 60.11 | 66.09 | 61.39 |
| MGUR (ICLR'24)[11] | 32.61 | 33.23 | 41.40 | 35.75 | 61.34 | 62.55 | **72.51** | 65.47 |
| SPIRIT (TOMM'24)[12] | 39.86 | **44.11** | 47.68 | 43.88 | 64.30 | 65.60 | 71.70 | 67.20 |
| **SADN (Ours)** | **40.01** | 43.67 | **48.04** | **43.91** | **65.10** | **66.05** | 70.93 | **67.36** |

$\hat{N}_i$ on the query features $Q_i$, which is formulated as:

$$\hat{Q}_i = (Q_i + \text{FC}(\hat{N}_i))/2, \qquad (11)$$

where FC is the fully-connected layer to transform the embeddings.

## 3.4 Similarity Measurement and Training Objectives

*3.4.1 Final Similarity.* Accordingly, the refined similarity score is computed as the cosine-similarity between the updated query features $\hat{Q}_i$ and candidate images, as:

$$\hat{s}_{ij} = \frac{\hat{Q}_i^\top \cdot T_j}{\|\hat{Q}_i\|\|T_j\|}. \qquad (12)$$

*3.4.2 Training Objectives.* Contrastive loss [3, 31, 38] is commonly used in the composed image retrieval task to enforce the query features to be close to positives and far away from negatives. However, the imbalances between positive and negative samples are severe when dealing with large batch sizes, which may arouse unstable convergences during training. Furthermore, models are inclined to attend to easy negative samples with comparatively low similarity scores for the given query, and the appropriate distances between the query and hard negative instances are not sufficiently improved in the metric learning. Instead, we adopt the improved focal loss to enhance the model's discrimination against hard negative samples. Specifically, the hard negatives are assigned with greater weights to increase the penalty on the high similarity scores between hard negative samples and the query samples. Hence, we design the ranking loss as:

$$\mathcal{L}_{rank} = \frac{1}{N} \sum_{i=1}^{N} -(1 - \frac{\tau \exp(s_{ii})}{\sum_{j=1}^{N} \tau \exp(s_{ij})})^\gamma \log(\frac{\tau \exp(s_{ii})}{\sum_{j=1}^{N} \tau \exp(s_{ij})}), \qquad (13)$$

where $\tau$ is a hyperparameter to allow the similarity score to adjust within certain limits. $\gamma$ is the modulating factor to adjust the strength of concentration on the hard negatives.

Finally, the overall optimization objective is the integration of the adaptation loss $\mathcal{L}_{adapt}$ and $\mathcal{L}_{rank}$ as follows:

$$\mathcal{L} = \mathcal{L}_{rank} + \lambda \mathcal{L}_{adapt}, \qquad (14)$$

where $\lambda$ weights the proportion of the adaptation loss $\mathcal{L}_{adapt}$ during the training process.

## 4 Experiments

### 4.1 Experimental Setup

*4.1.1 Datasets & Evaluation Metrics.* **FashionIQ** [42] is a fashion-domain dataset for composed image retrieval task. It contains 77,684 fashion images on three categories: dresses, shirts, and toptees. The candidate images for training, validating, and testing are divided into 46,609, 15,537, and 15,538 respectively. The partition ratio of composed queries in training, validation, and test sets is 3:1:1. We follow the original protocol for validating. Originating in NLVR$^2$ dataset [35], **CIRR** [29] includes 21,552 images and 36,554 correlated triplets on natural scenarios, with subsets consisting of visually similar pictures. The proportions of the triplets in training, validation, and test sets are 80%, 10%, and 10% respectively.

We utilize the commonly used evaluation metric Recall-rate at K (R@K) for the two datasets in composed image retrieval. Following [9, 13, 20], we set $K$ as 10 and 50 within dresses, toptees, and shirts in FashionIQ. For CIRR, apart from R@1, R@5, R@10, and R@50, we also report subset ranking results as Recall$_{subset}@K$.

*4.1.2 Implementation Details.* The visual and textual encoders initialized in CLIP [32] backbones are implemented as four ResNet-50 [19] models and Transformer frames respectively. We first fine-tune the visual and textual encoders for 10 epochs with the learning rate of $2 \times 10^{-6}$, and then freeze the parameters in encoders to train the proposed SADN models with the learning rate of $2 \times 10^{-5}$ and set the batch size as 2048. The default multi-modal compositor follows the Combiner [3] on the open-source community to obtain the initial similarities. We set the $K$ as 10 in neighbor representations in Sec. 3.2. The parameters $\alpha$ and $\beta$ are initialized as 0.5 and

**Table 2: Experiments results on CIRR. Best and second-best results are marked in bold and underlined respectively.**

| Methods | Recall@$K$ | | | | Recall$_{subset}$@$K$ | | | (R@5+Recall$_{subset}$@1)/2↑ |
|---|---|---|---|---|---|---|---|---|
| | K=1 | K=5 | K=10 | K=50 | K=1 | K=2 | K=3 | |
| TIRG (CVPR'19)[38] | 11.04 | 35.08 | 51.27 | 83.29 | 23.82 | 45.65 | 64.55 | 29.45 |
| MAAF (Arxiv'20)[14] | 10.31 | 33.03 | 48.30 | 80.06 | 21.05 | 41.91 | 61.60 | 27.04 |
| CIRPLANT (ICCV'21)[29] | 15.18 | 43.36 | 60.48 | 87.64 | 33.81 | 56.99 | 75.40 | 38.59 |
| ARTEMIS (ICLR'22)[13] | 16.96 | 46.10 | 61.31 | 87.73 | 39.99 | 62.20 | 75.67 | 43.05 |
| CLIP4Cir (CVPR'22)[3] | 33.59 | 65.35 | 77.35 | 95.21 | 62.39 | 81.81 | 92.02 | 63.87 |
| Chen *et al.* (Arxiv'23)[8] | 32.24 | 66.63 | 79.23 | 96.43 | 61.25 | 81.33 | 92.02 | 63.94 |
| BLIP4CIR (WACV'24)[30] | 40.17 | 71.81 | 83.18 | 95.69 | 72.34 | 88.70 | 95.23 | 72.07 |
| **SADN (Ours)** | **44.27** | **78.10** | **87.71** | **97.89** | **72.71** | **89.33** | **95.38** | **75.41** |

**Table 3: Ablation experiments on model designs.**

| Models | Dress | Shirt | Toptee | Mean |
|---|---|---|---|---|
| SADN w/o Neighbors | 38.37 | 41.28 | 45.57 | 41.74 |
| SADN w/o CIA | 39.71 | **43.81** | 47.52 | 43.68 |
| SADN w/o DBC | 39.37 | 43.23 | 47.73 | 43.44 |
| SADN w/o Residual Updating | 38.02 | 42.81 | 45.84 | 42.22 |
| SADN | **40.01** | 43.67 | **48.04** | **43.91** |

0.5 in Eq. 9. We fix the $\gamma$ in Eq. 13 as 2 empirically and $\tau$ as 100 for adjustment. The parameter $\lambda$ in Eq. 14 is fixed as 100 and more analysis would be found in Sec 4.4.1. All experiments are trained for 100 epochs on a single NVIDIA GeForce RTX 3090 Ti GPU via Pytorch, and achieve convergences within 5 hours with the memory cost of 63.12M.

## 4.2 Quantitative Experimental Results

*4.2.1 Results on FashionIQ.* The comparison of experimental results on FashionIQ dataset shown in Table 1 demonstrates the overall competitiveness of the proposed SADN. The upper column displays conventional methods without CLIP, and the middle column shows methods on the pretrained CLIP. The improvement of the proposed SADN is ascribed to the incorporation of neighbor representations on the query features, which distills semantic alignments from the target domain to enhance the salient correlations in the query. Besides, when comparing our model with MGUR [11] which is devoted to modeling the uncertainty learning in this coarsegrained retrieval task, a consistent recall rate growth could be observed. The growing recall rates implicate the neighborhood construction in SADN could facilitate the generalization and correlation extractions rather than sampling from Gaussian distributions.

*4.2.2 Results on CIRR.* Table 2 reports quantitative comparison results with other approaches on CIRR dataset. In comparison with BLIP4CIR [30] requiring high computation complexity on fine-tuning BLIP models, our proposed SADN still demonstrates promising retrieval results in a concise architecture by the improvement of 4.1% on the R@1. This comparison implies that the proposed SADN could effectively capture the latent alignments across the query and target to avoid the consumption of huge computation resources. A more remarkable improvement could be seen on R@50 mainly

due to the correspondence-induced adaption to strengthen the correlations from potential false negative samples. The gain on matching results about subsets demonstrates SADN is discriminatory while ranking visually similar images with subtle distinctions.

**Table 4: Ablation experiments on the different compositors.**

| Models | FashionIQ | | CIRR | |
|---|---|---|---|---|
| | R@10 | R@50 | R@5 | R$_{sub}$@1 |
| Image Only | 6.87 | 14.09 | 30.73 | 20.88 |
| Image Only + SADN | 7.12 | 16.25 | 31.26 | 21.41 |
| Text Only | 21.59 | 41.48 | 54.63 | 70.70 |
| Text Only + SADN | 24.35 | 45.52 | 56.59 | 74.72 |
| Summarization | 36.11 | 61.65 | 71.86 | 65.38 |
| Summarization+ SADN | 38.83 | 63.18 | 74.61 | 69.50 |
| TIRG$^{\dagger}$ [38] | 31.94 | 57.42 | 62.11 | 56.82 |
| TIRG$^{\dagger}$ [38]+ SADN | 33.25 | 58.68 | 63.88 | 57.71 |
| Combiner$^{\dagger}$ [3] | 37.89 | 65.95 | 76.82 | 71.47 |
| Combiner + SADN | 43.91 | 67.36 | 78.10 | 72.71 |

## 4.3 Ablation Studies

*4.3.1 Ablation Study of Effective Components.* To evaluate the impacts of each component in our SADN, Table 3 presents the R@10 results of ablation designs on the testbed of FashionIQ. "CIA" and "DBC" are the abbreviations for Correction-Induced Adaption and Divergence-based Correction respectively. We come to the following observations from Table 3: (1) Neighborhood construction is essential for attaining robust retrieval performance, which is confirmed by the recall rates decline when comparing the model without neighborhood to SADN. In this case, the unrelated candidate images prevail and dominate the distilling process when aggregating on the query features and disturb both the correspondence learning and the divergence measurement. (2) Both DBC and CIA could facilitate matching the positive target images and DBC is more effective than CIA as the improvements on the mean score shows. The whole SADN achieves the optimal retrieval performance, which suggests the distance measurement is significant to regulating the semantic distillation from the neighborhoods and suppressing the irrelevant components to aggregate on the query. Note that

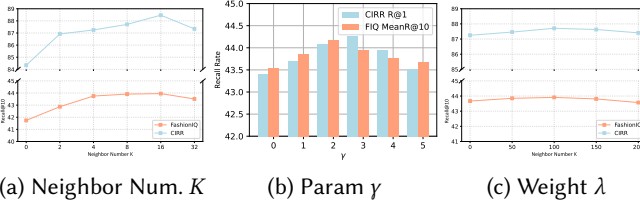

(a) Neighbor Num. $K$  (b) Param $\gamma$  (c) Weight $\lambda$

**Figure 3: Parameter sensitivity analysis of the neighbor number $K$, parameters $\gamma$ and $\lambda$.**

the shirt category has more visually similar samples due to nearly identical outlines of shirts, making model discrimination in DBC more essential. (3) Residual connection from the neighborhood in the target domain on the query is effective in assimilating more distribution characteristics from the target domain into the hybrid-modal domain, as the comparison between the last two rows show.

*4.3.2 Ablation Study of Different Compositors.* To test the effectiveness of the proposed architecture, we also report results on different backbones for the compositor in Table 4, where † means re-implementation. The proposed SADN could serve as a plug-and-play design to be integrated into different models for further improving the matching performance. The unsatisfactory performances on the original "Image Only" and "Text Only" indicate incomplete query and insufficient fusion for the query compositor would undermine the query-to-target matching process. The overall enhancement after aggregation demonstrates the generalization of SADN.

**Table 5: Ablation experiments on the training objectives.**

| Models | Dress | Shirt | Toptee | Mean |
|---|---|---|---|---|
| SADN w/o $\mathcal{L}_{adapt}$ | 39.51 | **43.82** | 47.67 | 43.67 |
| SADN w/o $\mathcal{L}_{rank}$ | 38.37 | 41.94 | 45.84 | 42.15 |
| SADN w $\mathcal{L}_{BCE}$ | 39.32 | 43.76 | 47.53 | 43.54 |
| SADN w $\mathcal{L}_{triplet}$ | 38.02 | 41.46 | 45.64 | 41.71 |
| SADN | **40.01** | 43.67 | **48.04** | **43.91** |

*4.3.3 Ablation Study of Losses.* To estimate the impacts of the losses on the retrieval performance, Table 5 shows the ablative experiments with various training objectives on R@10 metric. $\mathcal{L}_{adapt}$ is indispensable for the robustness to guide the correlation learning under the guidance of the target features. After joining the $\mathcal{L}_{rank}$, the recall rates further increase as the reweight factors address the imbalance partition of the positive and negative samples. The results from the "SADN w $\mathcal{L}_{BCE}$" and "SADN w $\mathcal{L}_{triplet}$" are inferior to the proposed SADN for the cross-entropy treats all the negatives equally regardless of false negatives and hard negatives, and the model equipped with triplet loss might be subjected to the convergence difficulty and high sensitivity to the noisy samples.

## 4.4 Further Analysis

*4.4.1 Analysis on Parameter Sensitivity.* We further investigate the parameter sensitivity of $K$, $\gamma$, and $\lambda$ in neighborhood construction,

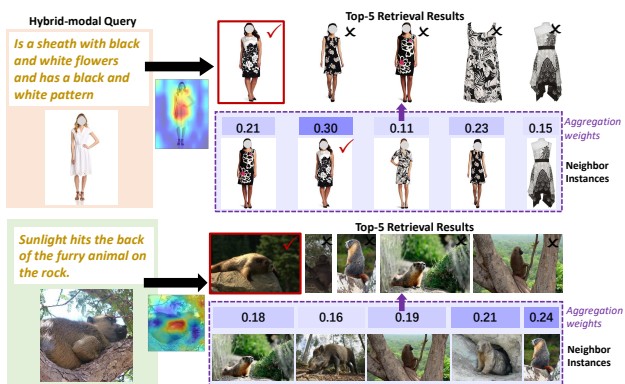

**Figure 4: Retrieval results of SADN on FashionIQ and CIRR.**

Eq. 13 and Eq. 14 on FashionIQ and CIRR. As shown in Figure 3 (a), the recall rates first increase and then flow with the growth of the number of $K$, since the limited neighbor representations restrict the semantic extraction from the target domain while the excess neighbors bring redundancy and noise. In terms of the $\gamma$ to manage the modulating factor, the optimal results occur at the $\gamma$ set as 2 for FashionIQ and 3 for CIRR. The difference in the optimal settings of $\gamma$ may come from the probability of the hard negative samples on CIRR being higher than FashionIQ, which requires more concentration on these samples during training. The parameter $\lambda$ controls the contribution of the $\mathcal{L}_{adapt}$ when optimizing SADN. An obvious drop could be observed when it is fixed as 0 and the advisable value ranges from 50 to 150 to cooperate with $\mathcal{L}_{rank}$ to promote the semantic distillation and discrimination.

*4.4.2 Visualization Results.* We also show the visualization retrieval results with neighbor images and aggregation weights in Figure 4. The heatmaps generated by GradCAM [34] on the last convolutional layer of the CLIP visual encoders demonstrate that our SADN could attend to the areas that the modifiers require. The target images (in red boxes) rank at the top through dynamically aggregating the neighbor features based on the semantic relations and divergence measurements in SADN.

## 5 Conclusion

In this paper, we proposed a novel semantic distillation from neighborhood dubbed SADN for composed image retrieval. To mitigate the heterogeneous gap between the query and target representations, we first constructed a neighborhood from the target domain for each query and introduced the semantic aware aggregation to refine the query features with dynamic weights. Specifically, we applied correlation-induced adaption to capture shared semantics from the neighbor features and divergence-based correction to restrain the irrelevant integration of noise. Experimental results and ablation studies verified the overall superiority and generalization of the proposed SADN. In the future, we would further investigate the feasibility of semantic-aware neighborhood construction on other multi-modal tasks.

# Acknowledgments

This work was supported by the National Key R&D Program of China (2022YFB4701400/4701402), SSTIC Grant (KJZD202309231151 06012, KJZD20230923114916032), and Beijing Key Lab of Networked Multimedia.

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
