# OpenReview forum: "Semantic Distillation from Neighborhood for Composed Image Retrieval"
_acmmm.org/ACMMM/2024/Conference — MM2024 Poster_

### Official Review · Reviewer_7YLn · 2024-05-14

**Rating:** 5
**Confidence:** 3

**Summary:**

Objective of composed image retrieval. The idea is to give as input an image and a text modifier and to get the corresponding image.
The idea is to propose two modules focusing on either converging features or diverging features to ensure a better adequation between the domain of input features and that of the target features.

The paper deals with the task of composed image retrieval. More specifically, it is the task of retrieving an image similar to an image modified by a text. The objective of this task can be apprehended as a domain adaptation problem, to make sure that features jointly extracted from the query image and the textual modifier correspond to the features of the target image. To this end, the proposed approach introduces two modules aiming towards modifying the query feature for the final retrieval in the target domain to better perform.

**Strengths:**

First of all, the paper is well-written even if a bit dense. The reason why the problem is interesting is well presented, especially the fact that the annotation process is often biased towards dissimilarity. The positioning of the approach is clear and the different types of existing solutions are well defined.

The problem is well defined and the presentation of the proposed solution is extensive and clear (if a bit dense), especially with the clear overview of Figure 2. The evaluation is extensive against many methods and the improvement is very clear. The detailed ablation study helps prove the impact of all proposed modules.

Points must also be given for the quality of the supplementary material. It is informative and gives details in all aspects of the paper. Furthermore, the presence of the code at submission time is greatly appreciated.

**Limitations:**

Although the quality of the paper is undeniable, some aspects should be addressed in my opinion.

First of all, even if the section 3 is clear and detailed, it remains quite dense to apprehend in one pass. It could be interesting to maybe have a diagram showing the different sets that are being compared, it would help understand the textual description. Furthermore, there seems to be an incoherence at the end of section 3.2 when the set $S_N^j$ is mentioned rather than $S_i^N$.

In the evaluation part, there is first an incoherence at the end of section 4.3.1. Isn’t it the last two lines rather than the last two columns of table 3 that should be compared ? Furthermore, it should be interesting to maybe underline the second best per column in table 1 and 2 because it seems that the proposed approach is mainly the best whereas the second best is divided between many methods. A question also arose regarding the parameter sensitivity, would it change with other datasets ? For instance with many more classes ? Or a nested classification ?

Finally, some information in terms of training time would be appreciated to compare with other methods.

Typos :
- l. 150-151 : “negative examples” but “is”
- l. 380 :  isn’t it the j-th target ?
- Supp. Mat : Section 4 title

**Suitability:**

3

---

### Official Review · Reviewer_W5wr · 2024-05-17

**Rating:** 4
**Confidence:** 4

**Summary:**

This paper proposes a semantic distillation from neighborhood called SADN for composed image retrieval. It first constructed a neighborhood from the target domain for each query and introduced the semantic aware aggregation to refine the query features with dynamic weights. The semantic aware aggregation includes a correlation-induced adaption to capture shared semantics and divergence-based correction to restrain the irrelevant integration of noise. Experimental results verified the superiority and generalization on different backbones of the proposed SADN.

**Strengths:**

1.	This paper is well-written with good readability.

2.	Extensive experiments and ablation studies are conducted to demonstrate the superiority and generalization of the proposed SADN.

**Limitations:**

1.	Why the designed Correspondence-Induced Adaption can distill correlated semantics from false negatives in the neighborhood? The authors should provide more in-depth discussions and analysis.

2.	What is the setting of baseline in Table 3? Does it mean that SADN w/o Neighbor-Aware Aggregation in Figure 2, and directly use $Q_i$ for similarity ranking? If yes, comparing the baseline with the compared methods in Table 1, why this simple baseline can outperform all of them? Do the baselines in Table 1 have the same backbones for image and text modality with SADN?

3.	The kernel function at Line 378 is not correct. It should be $g(T_j, \Psi)$.

**Suitability:**

3

---

### Official Review · Reviewer_PGLe · 2024-05-24

**Rating:** 2
**Confidence:** 4

**Summary:**

This paper investigates the composed image retrieval task, which aims to address the distribution gap between the multimodal query combination and the target image. To address this problem, this paper proposes SemAntic Distillation from Neighborhood (SADN), which constructs neighbourhood samples for each query from the target domain and further aggregates neighbourhood features using adaptive weights to reconstruct the query representation. Experiments show that the neighbourhood semantic distillation proposed in this paper significantly outperforms existing baselines and achieves better retrieval accuracy. However, this paper still has some obvious problems in terms of methodology discourse and experiments.

**Strengths:**

a.This paper proposes a novel neighbourhood-based semantic distillation method (SADN) for combinatorial image retrieval. The method integrates semantic features from the target image into the multimodal query representation, thereby bridging the distributional differences between the multimodal query and the target image.
b.The framework proposed in this paper constructs a neighbourhood for each multimodal query and captures the inherent semantic relevance. This is achieved by filtering out irrelevant parts from the target domain and reconstructing the multimodal query features using adaptive weighting, thus improving the retrieval performance of the model.
c.This paper adaptively align the multimodal query and the target image under the guidance of the target image and optimise the similarity ranking using divergence metric.

**Limitations:**

a. The integration of related works in this paper is not exhaustive, and some of them with significant relevance to this paper are not analysed as they should be. For example, TG-CIR [1] investigated similarity-guided candidate image ranking, which is analogous to the motivation of this paper. However, the authors did not fully explain the differences between this paper and previous similar works.
b. The experimental part of this paper adopted CLIP as the backbone, but only a few of the selected baselines adopted the same backbone. Furthermore, their performance was not as advanced as that of existing SOTA. In fact, there are many state-of-the-art works (e.g., TG-CIR [1], FAME-ViL [2], and SPIRIT [3]), have adopted CLIP as the backbone. However, the authors have not yet compared their model with these other works, which makes it challenging to demonstrate the superiority of the model proposed in this paper. It is recommended that the authors include comparisons with more advanced backbones. Furthermore, the authors do not explicitly state the specific version of CLIP that was employed, and it is recommended that this information be provided.
c. The analysis of experimental results in this paper requires further strengthening. For instance, in the ablation experiments, the paper does not analyse the anomalous experimental results of FashionIQ-Shirt in P7-L729 and P8-L833.
Reference：
[1] Wen H, Zhang X, Song X, et al. Target-guided composed image retrieval[C]//Proceedings of the 31st ACM International Conference on Multimedia. 2023: 915-923.
[2] Han X, Zhu X, Yu L, et al. Fame-vil: Multi-tasking vision-language model for heterogeneous fashion tasks[C]//Proceedings of the IEEE/CVF Conference on Computer Vision and Pattern Recognition. 2023: 2669-2680.
[3] Chen Y, Zhou J, Peng Y. SPIRIT: Style-guided Patch Interaction for Fashion Image Retrieval with Text Feedback[J]. ACM Transactions on Multimedia Computing, Communications and Applications, 2024.

**Suitability:**

3

---

### Official Review · Reviewer_asLN · 2024-05-24

**Rating:** 4
**Confidence:** 4

**Summary:**

This paper proposes a new SemAntic Distillation from Neighborhood (SADN) approach for composed image retrieval, which incorporates the shared semantic characteristics from the targets into query representations to bridge the distribution discrepancy between queries and targets. Specifically, the neighborhood of each query is constructed to capture inherent semantic correlations and filter irrelevant components from the target domain to reconstruct the query features with adaptive weights, which could further improve the robustness of the model. Correspondence-induced adaption with the guidance of the target representations is designed to distill alignments between query and target, in collaboration with divergences measurement to evaluate the distribution disparity for refining similarity ranking. Extensive experiments on public datasets demonstrate the effectiveness of the proposed method.

**Strengths:**

-	The authors have fully analyzed the issues of existing researches on composed image retrieval, including the distribution discrepancy between the distributions of query and target, the unlabeled positives instances in candidate set, and the unexplored subtle semantic differences.
-	The proposed method, Semantic Distillation from Neighborhood (SADN), incorporates the shared semantic characters from the targets into query representation to bridge the distribution discrepancy between targets. The idea is reasonable and novel to some extent.
-	The neighborhood construction for query captures inherent correlations and filters irrelevant components from target domain to reconstruct the query with adaptive weights, which is proved beneficial for improving the effectiveness of the model.
-	The extensive experiments prove the effectiveness of the proposed method.

**Limitations:**

-	The experimental results (Combiner + SADN) in Tab. 4 are a little different from those in Tab. 1 and 2. Is there any modification in method or network architectures while evaluating in Tab. 4 and Tab. 1,2.
-	The Fig. 2 is too complex to understand. The input of the framework and the loss functions are not clear indicated in the figure. The directions of the arrows in the figure are also confusing. The symbols in the figure are also in different form from those in the main body of the paper, such as C_i, D_i, and A_i are in capital form in Fig. 2, while in lowercase in the main body.
-	The time complexity and memory cost during training and testing should be analyzed in the paper, since real-time and lightweight retrieval is crucial for composed image retrieval in real applications. It seems two rounds of retrieval is essential for the proposed method.

**Suitability:**

3

---

### Meta-Review · Area_Chair_v599 · 2024-06-25

**Recommendation:** Accept (Poster)
**Confidence:** 5

**Metareview:**

This paper has been deliberated and the final decision is converged to be accepted due to the following recommendable facets:
1) Extensive investigation against existing methods kicking around and thus to ensure the proposed method stands out as its own extraordinary.
2) The proposed method is ideated and developed with high recognition as a reasonable, viable and more intuitive framework.
3) Sufficient experiments are conducted to demonstrate/prove the effectiveness of the proposed method.

Though this paper is praiseworthy, reviewers have spotted several limitations which will turn out to benefit the paper if those can be pulled off.
1) prominent query is the experimental results which are inconsistent to be reported across two tables.
2) computational overheads and memory costs should be provided.